# Inverse Transform Using Linearity for Video Coding

Hyeonju Song and Yung-Lyul Lee *

Department of Computer Engineering, Sejong University, 209 Neungdong-ro, Gwangjin-gu, Seoul 05006, Korea; hjsong@sju.ac.kr
* Correspondence: yllee@sejong.ac.kr; Tel.: +82-2-3408-3753

**Abstract:** In hybrid block-based video coding, transform plays an important role in energy compaction. Transform coding converts residual data in the spatial domain into frequency domain data, thereby concentrating energy in a lower frequency band. In VVC (versatile video coding), the primary transform is performed using DCT-II (discrete cosine transform type 2), DST-VII (discrete sine transform type 7), and DCT-VIII (discrete cosine transform type 8). Considering that DCT-II, DST-VII, and DCT-VIII are all linear transforms, inverse transform is proposed to reduce the number of computations by using the linearity of transform. When the proposed inverse transform using linearity is applied to the VVC encoder and decoder, run-time savings can be achieved without decreasing the coding performance relative to the VVC decoder. It is shown that, under VVC common-test conditions (CTC), average decoding time savings values of 4% and 10% are achieved for all intra (AI) and random access (RA) configurations, respectively.

**Keywords:** VVC (versatile video coding); HEVC (high efficiency video coding); linear inverse transform; computational complexity; DCT (discrete cosine transform); discrete sine transform (DST); BD-rate

---

## 1. Introduction

Transform coding is an important part of video coding, and it has been successfully adopted in most video coding standards, including MPEG-4 AVC (advanced video coding)/H.264 [1,2], HEVC (high efficiency video coding)/H.265 [3,4], and VVC/H.266 [5,6]. In hybrid block-based video coding, a transform is applied to the residual signal obtained after intra/inter-prediction, and the residual signals in the spatial domain are converted into the frequency domain. An efficient transform can concentrate more energy in the low-frequency components in the frequency domain. It is well known that the Karhunen–Loève transform (KLT) [7] is the optimal transform in terms of data decorrelation and compaction. However, the KLT is not used in actual transform coding because of its high complexity, as it computes the eigenvectors corresponding to a signal-dependent covariance matrix, and there is no fast computation algorithm in general. Since DCT-II provides good approximations of the KLT under the first-order Markov condition, many video coding standards use DCT-II [8] instead of the KLT. However, because of the diverse characteristics of images and videos, DCT-II is not always an optimal transform in terms of energy compaction and decorrelation. To solve this problem, alternative transform schemes such as DCT-II/DST-VIII for video coding [9] and enhanced multiple transform (EMT) [10] have been proposed. Moreover, with the first-order Gauss–Markov model for image signals, it is mathematically proven in [11] that DST-VII approximates the KLT well. Thus, HEVC specified the core transform based on DCT-II for the $4 \times 4$, $8 \times 8$, $16 \times 16$, and $32 \times 32$ predicted residual blocks and an alternate transform based on DST-VII for the $4 \times 4$ intra-predicted residual block [12].

Recently, with the substantial increase in the demand for high-definition and high-resolution videos, along with the growth of services such as video streaming, video compression technology with higher efficiency is required. The state-of-the-art video coding

---

standard—VVC—was developed by the Joint Video Experts Team (JVET) of the ITU-T WP3/16 Video Coding Experts Group (VCEG) and the ISO/IEC JTC 1/SC 29/WG11 Moving Picture Experts Group (MPEG), and it was finalized in July 2020. VVC was designed to meet various media needs such as UHD (ultra high definition) video, HDR (high dynamic range) video, WCG (wide color gamut) video, screen content video, and 360° video. During the development of the VVC standard, the joint separable and non-separable transforms [13] were proposed to improve the efficiency of transform. EMT using the separable property of transforms selects the predefined horizontal and vertical transforms, or the best horizontal and vertical transforms, in terms of coding efficiency from DCT-II, DCT-V, DCT-VIII, DST-I, and DST-VII [8]. Moreover, a non-separable secondary transform (NSST) [14] is proposed to operate with EMT, in which the NSST is applied as a secondary transform after EMT. Finally, both a simplified EMT and the NSST are adopted in the VVC standard.

In VVC, transform coding is largely divided into two processes: the primary transform and the secondary transform. The primary transform is a method used in traditional video coding standards. The simplified EMT that is applied to the predicted residual signals is used as the primary transform of VVC with the name multiple transform selection (MTS). In addition to DCT-II, DST-VII and DCT-VIII are also used as the transforms in MTS [15]. However, DST-VII and DCT-VIII are only applicable to luma blocks. Compared to HEVC, the maximum transform size is increased to $64 \times 64$. DCT-II is applied to transform block sizes from $4 \times 4$ to $64 \times 64$, while DST-VII and DCT-VIII are applied to transform block sizes from $4 \times 4$ to $32 \times 32$. Although large block-size transform is useful for higher resolution videos, it also increases computational complexity. To solve this problem, high frequency transform coefficients are zeroed out for large size transform blocks [16]. For 64-point DCT-II and 32-point DST-VII/DCT-VII, only the first 32 and 16 low-frequency coefficients are kept, respectively, and high frequency coefficients are zeroed out; this is also done in the last coefficient position coding and coefficient group scanning [15].

The secondary transform refers to an additional transform process that follows the primary transform. Low-frequency non-separable transform (LFNST) [17,18] is newly adopted in VVC and is applied to the top-left low-frequency region (ROI) of the primary transformed coefficients. When the LFNST is applied, all primary transform coefficients excluding ROI are zeroed out [19,20], and the output of LFNST is further quantized and entropy-coded [21]. In this paper, we analyze the number of multiplications of the existing fast transform methods in the VVC standard, and we propose a new fast inverse transform using the number of non-zero coefficients based on linearity to reduce the number of multiplications.

The rest of this paper is organized as follows. In Section 2.1, we introduce the transforms used in VVC, and we propose a fast inverse transform using linearity to reduce the computational complexity in Section 2.2. Section 3 discusses the experimental results. Finally, conclusions are given in Section 4.

## 2. VVC Transforms and Proposed Method

### 2.1. Introduction to DCT-II, DST-VII, and DCT-VIII

The 1D (one-dimensional) N-point transform and its inverse transform are respectively defined in Equations (1) and (2) as follows:

$$F(u) = \sum_{x=0}^{N-1} p(x) v_{u,x}, \quad u = 0, 1, 2, \ldots, N-1 \tag{1}$$

$$p(x) = \sum_{u=0}^{N-1} F(u) v_{u,x}, \quad x = 0, 1, 2, \ldots, N-1 \tag{2}$$

where $F(u)$ is the $N$-point transformed signal, $p(x)$ is its original signal, and $v_{u,x}$ is the basis element of $N \times 1$ $\mathbf{v_u}$ basis vector for each $u$, where $u,x = (0, 1, 2, \ldots, N-1)$, in DCT-II,

DST-VII, and DCT-IIII. $v_{u,x}$ for DCT-II, DST-VII, and DCT-VIII are respectively defined in Equations (3)–(5) as follows:

$$v_{u,x} = \alpha(u) \cos\left(\frac{u(2x+1)\pi}{2N}\right), \alpha(u) = \left\{ \begin{array}{ll} \sqrt{1/N}, & u = 0 \\ \sqrt{2/N}, & u = 1, 2, .., N-1 \end{array} \right., \tag{3}$$

$$v_{u,x} = \sqrt{\frac{4}{2N+1}} \sin\left(\frac{(2u+1)(x+1)\pi}{2N+1}\right) \tag{4}$$

$$v_{u,x} = \sqrt{\frac{4}{2N+1}} \cos\left(\frac{(2u+1)(x+1)\pi}{4N+2}\right) \tag{5}$$

*2.2. Propose Fast Inverse Transform Using Linearity*

The proposed inverse transform using a separable linear property is presented with the aim of reducing computational complexity. The proposed method focused on the primary inverse transform in the encoder and the decoder. At the inverse transform stage in the encoder and the decoder, the de-quantized transform coefficients after LFNST are input to a two-dimensional (2D) inverse transform. In most video coding standards, to reduce computational complexity, the 2D transform and inverse transform are implemented as separable transforms by applying the 1D inverse transform in Equation (2) to each row and each column. The separable inverse transform for non-square block size is expressed in Equation (6).

$$X' = B^T Y A \tag{6}$$

where $X'$ is the (n × m) inverse-transformed block, $Y$ is the (n × m) de-quantized transform block, $A$ is the (m × m) transform block, and $B^T$ is the (n × n) transform block, where n and m are the height and width of the block, respectively. Through the quantization and de-quantization processes, most of the transform coefficients become zero when the quantization value is high. If $Y$ consists of $N$ non-zero coefficients, $Y$ can be expressed as the sum of $N$ sub-blocks of the same size as $Y$ having only one non-zero coefficient, as shown in Equation (7), where $y_i$ is the $i$-th sub-block of $Y$.

$$Y = y_0 + y_1 + \ldots y_{N-1} \tag{7}$$

Figure 1 shows an example representing a 4 × 4 block composed of three non-zero coefficients that have been prepared using Equation (7). The DCT-II, DST-VII, and DCT-VIII have the following linear property expressed in Equation (8).

$$T(\alpha x + \beta y) = \alpha T(x) + \beta T(y) \tag{8}$$

where $T(\cdot)$ is the transform, $x$ and $y$ are the inputs of the transform, and $\alpha$ and $\beta$ are arbitrary scalar values. Using Equations (7) and (8), the inverse transform can be expressed as Equation (9).

$$X' = \sum_{k=0}^{N-1} B^T y_k A \tag{9}$$



**Figure 1.** Example of 4 × 4 block.

Assuming that a non-zero transform coefficient is in the $(i,j)$-th element in $Y$, which is named $y_l$, $B^T y_l A$, $0 \leq l \leq N - 1$, can be expressed as shown in Equation (10), using the basis vectors of transform $B^T$ and $A$.

$$B^T y_l A = \begin{bmatrix} \mathbf{v_0} & \cdots & \mathbf{v_{n-1}} \end{bmatrix} \begin{bmatrix} 0 & \cdots & 0 \\ \vdots & x_{i,j} & \vdots \\ 0 & \cdots & 0 \end{bmatrix} \begin{bmatrix} \mathbf{w_0}^T \\ \vdots \\ \mathbf{w_{m-1}}^T \end{bmatrix}, \text{ where } y_l = \begin{bmatrix} 0 & \cdots & 0 \\ \vdots & x_{i,j} & \vdots \\ 0 & \cdots & 0 \end{bmatrix} \quad (10)$$

where $x_{ij}$ is a non-zero transform coefficient in the $(i,j)$-th element in the $(n \times m)$ de-quantized transform block, $\mathbf{v_i}$ is the $i$-th basis vector of the transform $B^T$, and $\boldsymbol{w_i}$ is the $i$-th basis vector of the transform $A$. Equation (11) below is obtained by computing $B^T y_l$ from Equation (10).

$$B^T y_l = \begin{bmatrix} v_{0,0} & \cdots & v_{n-1,0} \\ \vdots & \ddots & \vdots \\ v_{0,n-1} & \cdots & v_{n-1,n-1} \end{bmatrix} \begin{bmatrix} 0 & \cdots & 0 \\ \vdots & x_{i,j} & \vdots \\ 0 & \cdots & 0 \end{bmatrix} = \begin{bmatrix} 0 \cdots & v_{i,0} * x_{i,j} & \cdots 0 \\ \vdots & v_{i,1} * x_{i,j} & \vdots \\ & \vdots & \\ 0 \cdots & v_{i,n-1} * x_{i,j} & \cdots 0 \end{bmatrix} \quad (11)$$

where $v_{i,j}$ is the $j$th element of the $i$-th basis vector. Finally, $B^T y_l A$ is obtained in a simplified form using Equation (12).

$$\begin{aligned} B^T y_l A &= \begin{bmatrix} 0 \cdots & v_{i,0} * x_{i,j} & \cdots 0 \\ \vdots & v_{i,1} * x_{i,j} & \vdots \\ & \vdots & \\ 0 \cdots & v_{i,n-1} * x_{i,j} & \cdots 0 \end{bmatrix} \begin{bmatrix} w_{0.0} & \cdots & w_{0,m-1} \\ \vdots & \ddots & \vdots \\ w_{m-1,0} & \cdots & w_{m-1,m-1} \end{bmatrix} \\ &= \begin{bmatrix} v_{i,0} * x_{i,j} * w_{j,0} & \cdots & v_{i,0} * x_{i,j} * w_{j,m-1} \\ \vdots & \ddots & \vdots \\ v_{i,n-1} * x_{i,j} * w_{j,0} & \cdots & v_{i,n-1} * x_{i,j} * w_{j,m-1} \end{bmatrix} = \begin{bmatrix} v_{i,0} * x_{i,j} * \boldsymbol{w_j^T} \\ \vdots \\ v_{i,n-1} * x_{i,j} * \boldsymbol{w_j^T} \end{bmatrix} \end{aligned} \quad (12)$$

When the proposed $(n \times m)$ inverse transform is applied for one non-zero coefficient, the number of multiplications becomes $n + (n \times m)$. Therefore, for an $(n \times m)$ transform block which has N non-zero coefficients, the total number of multiplications during the inverse transform using linearity is computed as expressed in Equation (13):

$$N \times (n + (n \times m)) \quad (13)$$

For an $(n \times m)$ transform block which has N non-zero coefficients, the total number of additions in the proposed inverse transform using linearity is computed in Equation (14), even though Equation (14) is not used in the proposed method:

$$(N - 1) \times (n \times m) \quad (14)$$

Therefore, the total number of multiplications in Equation (13) can be used for the fast inverse transform of the de-quantized transform block to reduce the computational complexity in the VVC inverse transform, only if the number of non-zero coefficients is small.

The determination of whether to perform the inverse transform using either the existing method with the separable property or the proposed method with the separable linear property is determined according to the threshold value that is obtained by comparing Equation (13) with the numbers of multiplications of DCT-II, DST-VII, and DCT-VIII in the VVC inverse transforms. The threshold is computed in advance as the maximum number of non-zero coefficients in the de-quantized transform block, wherein $N \times (n + (n \times m))$ does not exceed the number of multiplications in the VVC inverse transforms for every block size.

The proposed method proceeds as shown in Figure 2. First, the number of non-zero coefficients in the de-quantized transform block $Y$ is counted before the inverse transform process. Second, if the number of non-zero coefficients does not exceed the threshold, then the proposed inverse transform using separable linearity is performed; otherwise, the VVC inverse transform is performed.

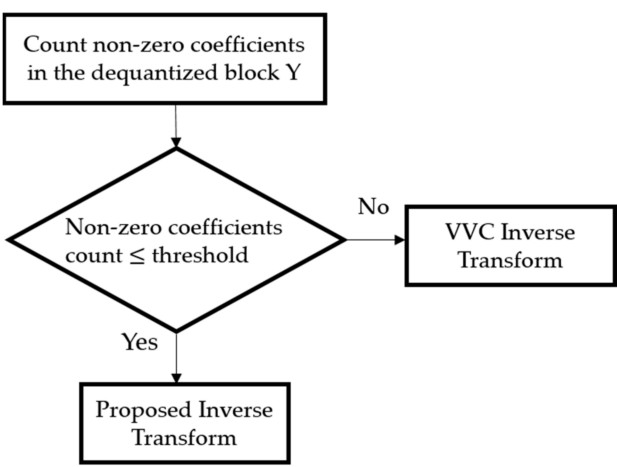

**Figure 2.** Flow chart of the proposed method.

The transform can be implemented using either the fast method or direct matrix multiplication. With direct matrix multiplication, the number of multiplications for 1D N-point transform is $N^2$. In VTM-8.2 (VVC Test Model 8.2) [22], DCT-II, DST-VII, and DCT-VIII are implemented using fast algorithms. For DCT-II, a fast algorithm using (anti-)symmetry properties of DCT-II is used. The even basis vectors of DCT-II are symmetric, while the odd basis vectors are anti-symmetric. Even–odd decomposition for the N-point input is performed as follows. The even and odd parts are calculated using the subset matrices obtained by the even and odd columns of the inverse transform matrix, respectively, and addition and subtraction operations are then performed between the even and odd parts to generate N-point output [23]. This fast method is also called a partial butterfly structure. Fast DST-VII and DCT-VIII with dual implementation support for VVC [24,25] were adopted as the primary transform solution. The fast method for DST-VII and DCT-VIII uses the inherited features of DST-VII and DCT-VIII to reduce the number of operations. In the DST-VII and DCT-VIII transform matrices, there are three features which are useful for reducing the number of computations [25]: first, N elements are included without considering sign changes. Second, only a subset of N elements is included without considering the sign changes. Third, except for zero, some transform vectors only contain a single element when neglecting the sign changes.

Table 1 lists the number of multiplications computed in the VTM-8.2 source code in each (n × m) block size, when the horizontal kernel and vertical kernel were both DCT-II. Table 2 presents the number of multiplications computed in the VTM-8.2 source code in each n × m block size, when the horizontal and vertical transforms were a combination of DST-VII and DCT-VIII; i.e., (DST-VII, DST-VII), (DST-VII, DCT-VIII), (DCT-VIII, DST-VII), and (DCT-VIII, DCT-VIII).

**Table 1.** Number of multiplications when the horizontal and vertical transforms are both DCT-II in the VTM-8.2.

| | | Width (m) | | | | | |
|---|---|---|---|---|---|---|---|
| | | **1** | **2** | **4** | **8** | **16** | **32** | **64** |
| | **1** | N/A | 2 | 8 | 24 | 88 | 344 | 684 |
| | **2** | 2 | 8 | 24 | 64 | 208 | 752 | 1432 |
| | **4** | 8 | 24 | 64 | 160 | 480 | 1632 | 2992 |
| **Height (n)** | **8** | 24 | 64 | 160 | 384 | 1088 | 3520 | 6240 |
| | **16** | 88 | 208 | 480 | 1088 | 2816 | 8320 | 13,760 |
| | **32** | 344 | 752 | 1632 | 3520 | 8320 | 22,016 | 32,896 |
| | **64** | 684 | 1496 | 3248 | 7008 | 16,576 | 43,904 | 65,664 |

**Table 2.** Number of multiplications when the horizontal and vertical transforms are a combination of DST-VII and DCT-VIII in the VTM-8.2.

| | | Width (m) | | | |
|---|---|---|---|---|---|
| | | **4** | **8** | **16** | **32** |
| | **4** | 64 | 320 | 636 | 2608 |
| **Height (n)** | **8** | 320 | 1024 | 2040 | 5984 |
| | **16** | 636 | 2040 | 4064 | 11,952 |
| | **32** | 2736 | 7008 | 13,984 | 29,760 |

As a reference, Table 3 lists the number of additions computed in the VTM-8.2 source code in each (n × m) block size, when the horizontal kernel and vertical transforms were both DCT-II. Table 4 presents the number of additions computed in the VTM-8.2 source code in each (n × m) block size, when the horizontal and vertical transforms were a combination of DST-VII and DCT-VIII. It can be easily computed by Equation (14) that the number of additions of the proposed method is smaller for all block sizes than those in VTM-8.2.

**Table 3.** The number of additions when the horizontal and vertical transforms are both DCT-II in the VTM-8.2.

| | | Width (m) | | | | | |
|---|---|---|---|---|---|---|---|
| | | **1** | **2** | **4** | **8** | **16** | **32** | **64** |
| | **1** | N/A | 2 | 8 | 28 | 100 | 372 | 802 |
| | **2** | 2 | 8 | 24 | 72 | 232 | 808 | 1668 |
| | **4** | 8 | 24 | 64 | 176 | 528 | 1744 | 3464 |
| **Height (n)** | **8** | 28 | 72 | 176 | 448 | 1248 | 3872 | 7312 |
| | **16** | 100 | 232 | 528 | 1248 | 3200 | 9152 | 16,032 |
| | **32** | 372 | 808 | 1744 | 3872 | 9152 | 23,808 | 37,568 |
| | **64** | 802 | 1732 | 3720 | 8208 | 19,232 | 49,472 | 76,992 |

**Table 4.** The number of additions when the horizontal and vertical transforms are combination of DST-VII and DCT-VIII in the VTM-8.2.

| | | Width (m) | | | |
|---|---|---|---|---|---|
| | | 4 | 8 | 16 | 32 |
| **Height (n)** | **4** | 88 | 344 | 796 | 3048 |
| | **8** | 344 | 1024 | 2264 | 6768 |
| | **16** | 796 | 2264 | 4960 | 13,968 |
| | **32** | 3224 | 7792 | 16,448 | 34,464 |

Table 5 lists the percentages of horizontal and vertical transforms in VTM-8.2 for all test sequences—which will be explained in Section 3—at each quantization parameter (QP) of 22, 27, 32, and 37, irrespective of the block sizes. As presented in Table 5, the fact that DCT-II/DCT-II takes up about 55.78% meant that the horizontal and vertical transforms were both DCT-II, and DST-VII/DCT-VIII taking up about 34.33% meant that the horizontal and vertical transforms were a combination of DST-VII and DCT-VIII; i.e., (DST-VII, DST-VII), (DST-VII, DCT-VIII), (DCT-VIII, DST-VII) and (DCT-VIII, DCT-VIII). Furthermore, other combinations taking up about 9.89% meant other transform combinations, except for DCT-II/DCT-II and DST-VII/DCT-VIII; i.e., the combinations of DCT-II/DST-VII and DCT-II/DCT-III. For the luma block, the transform pairs were mostly DCT-II/DCT-II or DST-VII/DCT-VIII. For the chroma block, since MTS was not enabled, only DCT-II/DCT-II was selected.

Since DST-VII/DCT-VIII requires more multiplications than DCT-II/DCT-II, when comparing Table 2 with Table 1, the threshold tables are divided into two types: DST-VII/DCT-VIII and DCT-II/DCT-II including other combinations. Other combinations are included with DCT-II/DCT-II to reduce the threshold tables, because the percentages of other combinations are lower than those of DCT-II/DCT-II and DST-VII/DCT-VIII, as presented in Table 5.

The threshold value in each n × m block—which is determined by comparing the number of multiplications in Tables 1 and 2 with the previously computed number of multiplications in Equation (8)—is determined by the combinations of the horizontal and vertical transforms presented in Tables 6 and 7, respectively. Table 6 shows the threshold value representing the number of non-zero coefficients in each n × m block, when the horizontal and vertical transforms were DCT-II/DCT-II or other combinations; Table 7 shows the threshold value representing the number of non-zero coefficients in each n × m block, when the horizontal and vertical kernels were a combination of DST-VII and DCT-VIII. For example, if the horizontal and vertical transforms were a combination of DST-VII/DCT-VIII and there were 14 or fewer non-zero coefficients shown in bold in the 8 × 8 block in Table 7, then the inverse transform was performed through the proposed inverse transform.

**Table 5.** Transform block percentage for each transform pair for the luma block.

| Class | QP | DCT-II/DCT-II | DST-VII/DCT-VIII (DST-VII, DST-VII), (DST-VII, DCT-VIII), (DCT-VIII, DST-VII), (DCT-VIII, DCT-VIII) | Other Combinations |
|---|---|---|---|---|
| A1 | 22 | 62.92% | 33.29% | 3.79% |
| | 27 | 67.21% | 30.02% | 2.77% |
| | 32 | 72.08% | 25.60% | 2.32% |
| | 37 | 77.10% | 20.79% | 2.11% |
| | Average | 69.83% | 27.43% | 2.75% |
| A2 | 22 | 65.48% | 34.02% | 1.4% |
| | 27 | 59.97% | 37.99% | 2.03% |
| | 32 | 60.65% | 37.55% | 1.80% |
| | 37 | 61.71% | 36.25% | 2.04% |
| | Average | 61.95% | 36.45% | 1.82% |
| B | 22 | 68.48% | 29.58% | 1.94% |
| | 27 | 59.57% | 36.17% | 4.26% |
| | 32 | 60.21% | 34.36% | 5.42% |
| | 37 | 64.46% | 30.15% | 5.39% |
| | Average | 63.18% | 32.57% | 4.25% |
| C | 22 | 59.38% | 33.70% | 6.91% |
| | 27 | 58.50% | 32.83% | 8.67% |
| | 32 | 59.02% | 31.35% | 9.63% |
| | 37 | 61.95% | 27.88% | 10.17% |
| | Average | 59.71% | 31.44% | 8.85% |
| D | 22 | 56.15% | 37.07% | 6.78% |
| | 27 | 53.47% | 37.21% | 9.32% |
| | 32 | 54.84% | 33.94% | 11.22% |
| | 37 | 58.66% | 29.10% | 12.24% |
| | Average | 55.78% | 34.33% | 9.89% |

**Table 6.** Threshold representing the number of non-zero coefficients in each block size, when the horizontal and vertical kernels are DCT-II/DCT-II or other combinations listed in Table 5.

| | | Width (m) | | | | | | |
|---|---|---|---|---|---|---|---|---|
| | | **1** | **2** | **4** | **8** | **16** | **32** | **64** |
| Height (n) | **1** | N\A | 1 | 2 | 3 | 5 | 10 | 10 |
| | **2** | 1 | 1 | 2 | 3 | 6 | 11 | 11 |
| | **4** | 2 | 2 | 3 | 4 | 7 | 12 | 11 |
| | **8** | 3 | 2 | 4 | 5 | 8 | 13 | 12 |
| | **16** | 5 | 4 | 6 | 7 | 10 | 15 | 13 |
| | **32** | 10 | 7 | 10 | 12 | 15 | 20 | 15 |
| | **64** | 10 | 7 | 10 | 12 | 15 | 20 | 15 |

**Table 7.** Threshold representing the number of non-zero coefficients in each block size, when the horizontal and vertical kernels are a combination of DST-VII and DCT-VIII (DST-VII/DCT-VIII).

| | | Width (m) | | | |
|---|---|---|---|---|---|
| | | **4** | **8** | **16** | **32** |
| | **4** | 3 | 8 | 9 | 19 |
| **Height (n)** | **8** | 8 | 14 | 15 | 22 |
| | **16** | 7 | 14 | 14 | 22 |
| | **32** | 17 | 24 | 25 | 28 |

## 3. Experimental Results

In this section, the coding performance of the inverse transform using linearity was compared with the VVC inverse transform. The proposed method using linearity implemented on top of VTM-8.2 was evaluated under the VVC CTC [26]. The sequences used in the test are summarized in Table 8 by class (resolution), frame count, frame rate, and bit depth. The experiments were conducted using the VVC reference software (SW), VTM-8.2, as an anchor under the all intra (AI) and random access (RA) configurations [26]. The tested quantization–parameter (QP) values were 22, 27, 32, and 37, respectively.

**Table 8.** Information on video sequences.

| Class | Sequence Name | Frame Count | Frame Rate | Bit Depth |
|---|---|---|---|---|
| A1 (4K) | Tango2 | 294 | 60 | 10 |
| | FoodMarket4 | 300 | 60 | 10 |
| | Campfire | 300 | 30 | 10 |
| A2 (4K) | CatRobot | 300 | 60 | 10 |
| | DaylightRoad2 | 300 | 60 | 10 |
| | ParkRunnung3 | 300 | 50 | 10 |
| B (1080p) | MarketPlace | 600 | 60 | 10 |
| | RitualDance | 600 | 60 | 10 |
| | Cactus | 500 | 50 | 8 |
| | BasketballDrive | 500 | 50 | 8 |
| | BQTerrace | 600 | 60 | 8 |
| C (WVGA) | RaceHorses | 300 | 30 | 8 |
| | BQMall | 600 | 60 | 8 |
| | PartyScene | 500 | 50 | 8 |
| | BasketballDrill | 500 | 50 | 8 |
| D (WQVA) | RaceHorses | 300 | 30 | 8 |
| | BQSquare | 600 | 60 | 8 |
| | BlowingBubbles | 500 | 50 | 8 |
| | BasketballPass | 500 | 50 | 8 |

Table 9 lists the average selection ratios of the proposed inverse transform and the inverse transform in the VVC reference SW for the Y, Cb, and Cr components that are decoded with the QP values of 22, 27, 32, and 37 under the RA configuration for the sequences of every class, where Y is a luma component and Cb and Cr are each chroma components. In Table 9, "VVC inverse transform" indicates the inverse transform used in the VTM-8.2 SW and "proposed" indicates the proposed linear transform, considering the threshold value. For the Y component, when the QP value was 22 for all class sequences, the average selection ratios of the VVC inverse transform and the proposed method were 65.71% and 25.96%, respectively; however, when the QP value was 37, the average selection ratios of the VVC inverse transform and the proposed method became 47.00% and 53.00%, respectively. Therefore, the average selection ratios of the proposed method for the Y

component gradually increased as the QP value was increased. Meanwhile, at the QP value of 22, although the average selection ratios of the VVC inverse transform and the proposed method were 55.78% and 44.22% for the Cb component, and 54.74% and 45.26% for the Cr component, those values respectively became 37.37% and 62.63% for the Cb component, and 37.81% and 62.19% for the Cr component at the QP value of 37. Therefore, in a similar manner to the Y component, the average selection ratios of the proposed method for the Cb and Cr components gradually increased as the QP value was increased. These results were as expected, because the higher the QP value is, the fewer non-zero coefficients there are in the quantization process. As presented in Table 9, the worst sequence, in which the proposed method was less selected, was the class D sequence. Even at the QP value of 37, the average selection ratio of the proposed method was 41.73% for the Y component. The best sequence, in which the proposed method was most selected, was the class A2 sequence from Table 9, where the average selection ratio of the proposed method was 63.02% at the QP value of 37 for the Y component.

**Table 9.** Average selection ratios of VVC inverse transform and the proposed method for each QP for all test sequences.

| QP | Class | Y | | Cb | | Cr | |
|----|-------|---|---|----|----|----|----|
| | | VVC Inverse Transform | Proposed | VVC Inverse Transform | Proposed | VVC Inverse Transform | Proposed |
| 22 | A1 | 56.84% | 43.16% | 44.38% | 55.62% | 33.62% | 66.38% |
| | A2 | 63.99% | 36.01% | 60.88% | 39.12% | 62.73% | 37.27% |
| | B | 66.76% | 33.24% | 51.27% | 48.73% | 50.71% | 49.29% |
| | C | 66.92% | 33.08% | 60.14% | 39.86% | 63.28% | 36.72% |
| | D | 74.04% | 25.96% | 62.21% | 37.79% | 63.38% | 36.62% |
| | Average | **65.71%** | **34.29%** | **55.78%** | **44.22%** | **54.74%** | **45.26%** |
| 27 | A1 | 43.20% | 56.80% | 42.60% | 57.40% | 31.22% | 68.78% |
| | A2 | 46.66% | 53.34% | 53.77% | 46.23% | 58.91% | 41.09% |
| | B | 54.20% | 45.80% | 43.86% | 56.14% | 45.41% | 54.59% |
| | C | 58.04% | 41.96% | 53.30% | 46.70% | 55.03% | 44.97% |
| | D | 66.22% | 33.78% | 54.34% | 45.66% | 56.23% | 43.77% |
| | Average | 53.66% | 46.34% | 49.57% | 50.43% | 49.36% | 50.64% |
| 32 | A1 | 41.50% | 58.50% | 39.82% | 60.18% | 27.06% | 72.94% |
| | A2 | 38.67% | 61.33% | 47.81% | 52.19% | 52.77% | 47.23% |
| | B | 48.24% | 51.76% | 38.22% | 61.78% | 40.61% | 59.39% |
| | C | 52.07% | 47.93% | 45.14% | 54.86% | 46.65% | 53.35% |
| | D | 61.53% | 38.47% | 45.39% | 54.61% | 48.38% | 51.62% |
| | Average | 48.40% | 51.60% | 43.28% | 56.72% | 43.09% | 56.91% |
| 37 | A1 | 43.63% | 56.37% | 33.00% | 67.00% | 22.67% | 77.33% |
| | A2 | 36.98% | **63.02%** | 43.11% | 56.89% | 47.07% | 52.93% |
| | B | 46.69% | 53.51% | 33.19% | 66.81% | 35.73% | 64.27% |
| | C | 49.65% | 50.35% | 38.41% | 61.59% | 41.10% | 58.90% |
| | D | 58.27% | **41.73%** | 39.13% | 60.87% | 42.51% | 57.49% |
| | Average | **47.00%** | **53.00%** | **37.37%** | **62.63%** | **37.81%** | **62.19%** |

Table 10 lists the comparison results of the coding performance and the computational complexity of the proposed method with the VVC inverse transform. The Bjøntegaard delta bitrates (BD-rate) [27,28] was used to evaluate coding performance. The negative BD-rates values indicated bit-saving of the proposed method compared to the VVC inverse transform in the same PSNR (pear signal-to-noise ratio) value. The runtime $\Delta T$ was calculated as the ratio of the proposed method's runtime on top of VTM-8.2, $T_{proposed}$, to VTM-8.2's runtime, $T_{VTM\text{-}8.2}$, as indicated in Equation (15). The total encoding and decoding runtime ratios

of the proposed approach on top of VTM-8.2 to VTM-8.2 are respectively represented in Table 10 as $\Delta EncT$ and $\Delta DecT$.

$$\Delta T = \frac{T_{Proposed}}{T_{VTM\text{-}8.2}} \times 100\% \tag{15}$$

**Table 10.** Experimental results of VVC inverse transform vs. the proposed method for class (A to D) sequences.

| Class | Sequence | BD-Rates (%) and Runtime Ratios in AI | | | | | BD-Rates (%) and Runtime Ratios in RA | | | | |
|---|---|---|---|---|---|---|---|---|---|---|---|
| | | Y | Cb | Cr | EncT | DecT | Y | Cb | Cr | EncT | DecT |
| A1 | Tango2 | 0.03% | −0.22% | 0.05% | 98% | 98% | 0.04% | −0.30% | 0.11% | 90% | 89% |
| | FoodMarket4 | −0.01% | 0.12% | 0.09% | 98% | 99% | 0.04% | −0.08% | −0.05% | 87% | 82% |
| | Campfire | 0.00% | −0.07% | −0.11% | 99% | 97% | 0.02% | −0.01% | 0.04% | 93% | 96% |
| | Average | 0.01% | −0.06% | 0.01% | 98% | 98% | 0.03% | −0.13% | 0.03% | 90% | **89%** |
| A2 | CatRobot | 0.01% | 0.00% | 0.08% | 98% | 94% | −0.02% | −0.11% | −0.12% | 95% | 87% |
| | DaylightRoad2 | 0.02% | −0.10% | 0.03% | 99% | 98% | −0.01% | −0.06% | 0.31% | 97% | 90% |
| | ParkRunnung3 | 0.00% | 0.00% | 0.00% | 98% | 95% | −0.01% | 0.01% | 0.02% | 94% | 86% |
| | Average | 0.01% | −0.04% | 0.04% | 98% | 96% | −0.01% | −0.05% | 0.07% | 95% | **88%** |
| B | MarketPlace | −0.01% | 0.07% | 0.05% | 88% | 95% | 0.01% | 0.03% | 0.14% | 99% | 84% |
| | RitualDance | −0.01% | −0.13% | 0.07% | 89% | 95% | 0.07% | 0.18% | 0.08% | 98% | 99% |
| | Cactus | 0.01% | −0.03% | 0.00% | 87% | 92% | 0.01% | −0.03% | −0.11% | 98% | 97% |
| | BasketballDrive | 0.04% | −0.12% | −0.03% | 88% | 97% | 0.06% | −0.21% | −0.02% | 99% | 93% |
| | BQTerrace | 0.00% | 0.15% | 0.15% | 87% | 91% | −0.01% | −0.05% | −0.39% | 95% | 96% |
| | Average | 0.01% | −0.01% | 0.05% | 88% | 94% | 0.03% | −0.01% | −0.06% | 98% | **94%** |
| C | RaceHorses | 0.01% | 0.10% | −0.05% | 99% | 94% | 0.02% | −0.25% | −0.03% | 98% | 77% |
| | BQMall | 0.00% | −0.13% | −0.16% | 100% | 92% | 0.07% | 0.05% | 0.15% | 98% | 86% |
| | PartyScene | 0.00% | 0.07% | −0.11% | 101% | 92% | 0.00% | −0.03% | −0.10% | 99% | 91% |
| | BasketballDrill | 0.00% | −0.04% | 0.06% | 99% | 98% | 0.01% | −0.03% | −0.13% | 99% | 94% |
| | Average | 0.00% | 0.00% | −0.06% | 100% | 94% | 0.02% | −0.07% | −0.03% | 99% | **87%** |
| D | RaceHorses | 0.02% | 0.00% | −0.27% | 98% | 98% | 0.01% | −0.07% | 0.18% | 99% | 95% |
| | BQSquare | −0.01% | 0.34% | 0.06% | 99% | 97% | 0.04% | 0.40% | 0.25% | 99% | 100% |
| | BlowingBubbles | 0.01% | −0.26% | −0.03% | 99% | 95% | 0.00% | 0.02% | −0.35% | 97% | 98% |
| | BasketballPass | −0.01% | −0.13% | 0.22% | 99% | 99% | 0.04% | 0.35% | −0.02% | 98% | 81% |
| | Average | 0.00% | −0.01% | −0.01% | 99% | 97% | 0.02% | 0.18% | 0.02% | 98% | **93%** |
| | Overall | **0.00%** | **−0.02%** | **0.01%** | **96%** | **96%** | **0.02%** | **−0.01%** | **0.00%** | **96%** | **90%** |

The proposed inverse transform with linearity was implemented in the encoder and decoder for the experiments. Because the separable transform in the VVC standard uses 16-bit precision after the vertical and horizontal transforms, encoder and decoder mismatches may occur if the proposed linear transform is solely applied to the decoder side.

In Table 10, the proposed method approximately maintains the average BD-rates of Y, Cb, and Cr by 0.00, −0.02, and 0.01%, respectively, with average encoding and decoding time reductions of approximately 4% and 4%, respectively, for classes (A to D) under the AI configuration. Furthermore, in the RA configuration, the proposed method reduced the average encoding and decoding times by approximately 4% and 10%, respectively, for classes (A to D), while maintaining average BD-rates of 0.02, −0.01, and 0.00% for Y, Cb, and Cr, respectively. The average decoding time was more reduced than the encoding time in the RA configuration, which was attributed to the fact that the VVC decoder is much simpler than that of the VVC encoder in terms of complexity. Finally, when the proposed inverse transform using linearity with a separable property was applied to the VVC encoder and decoder, it achieved run-time savings while maintaining coding performance compared to the VVC decoder.

Fast encoding methods only in the encoder side were proposed to reduce the encoding complexity of VVC, but all fast encoding methods increased BD-rates [29,30] in terms of the bit-rate reduction; thus, the proposed inverse transform using linearity in the decoder

side differs from these approaches, in that it keeps the BD-rate in VVC while reducing decoding complexity. If the proposed inverse transform was applied to the VVC standard, the inverse transform of the VVC standard should be changed to include the proposed method. Therefore, the proposed method can be considered in the next-generation video coding standards, as it reduces decoding complexity while the BD-rate is maintained.

## 4. Discussion

The previously proposed fast methods were mainly addressed to reduce complexity in the video encoder with BD-rate loss. In [29], a fast intra-mode decision algorithm was proposed, and the result showed the encoding time savings of 51~53% with BD-rate loss of 0.93~1.08%. A low-complexity CTU (coding tree unit) partition structure decision and fast intra-mode decision were proposed in [30], and showed average encoding time savings of 63% with a BD-rate loss of 1.93%. Fast encoders for video coding only reduce the encoder complexity, while BD-rates always increased without decreasing the decoder complexity. However, the proposed fast inverse transform is different from the fast encoders, in that it reduces the complexity in both the encoder and decoder while maintaining the BD-rate of the VVC standard. In the RA configuration, the proposed method reduces the average encoding and decoding times by approximately 4% and 10%, respectively, while maintaining average BD-rates.

If the proposed inverse transform using the number of non-zero coefficients is applied to the VVC standard, the inverse transform of the VVC standard should be changed to include the proposed method. However, the proposed method can be considered in next-generation video coding standards because it achieves decoding run-time saving, while maintaining average BD-rate. In addition to that, the proposed method is more effective at high QP values than at low QP values, because the higher the QP value is, the fewer non-zero coefficients there are.

## 5. Conclusions

To reduce computational complexity, this paper proposed an inverse transform using the number of non-zero coefficients based on linearity with separability. To reduce the number of multiplications in the inverse transform process, the proposed inverse transform makes use of the number of non-zero coefficients based on linearity. The experiment was conducted using VTM-8.2. Under the AI and RA configurations, the proposed inverse transform, on top of the VTM-8.2 SW, reduced decoding time by average values of 4% and 10%, respectively. We believe that the proposed inverse transform can be combined with the VTM-8.2 SW's existing transform implementation to provide a fast decoder for use in practice. The proposed method can be considered in next-generation video coding standards, as it reduces decoding complexity while the BD-rate is maintained.

**Author Contributions:** Conceptualization, Y.-L.L. and H.S.; methodology, Y.-L.L.; software, H.S.; validation, Y.-L.L. and H.S.; formal analysis, Y.-L.L. and H.S.; investigation, Y.-L.L. and H.S.; resources, Y.-L.L. and H.S.; writing, review and editing, Y.-L.L. and H.S.; supervision, Y.-L.L.; project administration, Y.-L.L.; funding acquisition, Y.-L.L. All authors have read and agreed to the published version of the manuscript.

**Funding:** This research was in part supported by the National Research Foundation of Korea (NRF) grant funded by the Korea government (NRF-2018R1D1A1B07045156).

**Conflicts of Interest:** The authors declare no conflict of interest.

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
