# Peer review of "Inverse Transform Using Linearity for Video Coding"

_electronics, doi:10.3390/electronics11050760_

Round 1

Reviewer 1 Report

The manuscript is well written, well organized, and clear. The results show a higher performance of the proposed computation of inverse transforms.

Additions also affect complexity. Please add information on the number of additions for different transform types and sizes in original and proposed variants.  

Reviewer 2 Report

The authors should accurately address the below comments.

  • Keywords: We suggest that the authors should replace keywords such as “video coding” and “transform” because these keywords are already found in the article title. It is better that they replace them with other keywords to increase the reach of the manuscript.
  • Introduction Section: The authors should add the main contributions briefly at the end of the introduction.
  • Methodology Section: It should be structured, sub-headings should be added to facilitate tracking and understanding.
  • Discussion Section: The authors should add a section where they discuss comparing their results with those of existing research. Also, they should clarify the limitations of the proposed method. There is a lot of recent research out there that can be used for comparison. As this discussion and comparison can clarify the fairness and rationality of the results of the proposed method.
  • Conclusions: Future directions should be added to the conclusion section.
  • Figures and Tables: All figures and tables are shown before they are used in the text.
  • References List: The list of references is recent, and all references are related to the research topic but it is not sufficient for this study. The names of the researchers must follow the style of the journal format. The double quotation should be omitted from the research titles in the list of references. Some search names in the reference list begin an uppercase letter for each word (such as [4], [5] ... etc.) and others use only an uppercase letter in the first word (such as [2], [9] … etc.), authors should standardize style. All journal names should be italic. Some references do not contain enough information such as [16], [18] … etc. Some links do not work in the reference list like [22] … etc. The list of references requires an extensive check.
  • English Writing: This article requires extensive proofreading. Authors should check the entire article to remove all extensive mistakes (grammatical and typos) and to improve English writing quality.

Round 2

Reviewer 2 Report

The authors should accurately address the below comments.

  • Introduction Section: This comment still requires a response. The authors should add the main contributions briefly at the end of the introduction.
  • Discussion Section: The authors did not respond accurately to this comment. The authors should add a section where they discuss comparing their results with those of existing research. Also, they should clarify the limitations of the proposed method.
  • Figures and Tables: All figures and tables are shown before they are used in the text.

Author Response

Attached please find the Q&A file.
